# The Role of Calix[n]arenes and Pillar[n]arenes in the Design of Silver Nanoparticles: Self-Assembly and Application

**DOI:** 10.3390/ijms21041425

**Published:** 2020-02-20

**Authors:** Pavel Padnya, Vladimir Gorbachuk, Ivan Stoikov

**Affiliations:** A.M. Butlerov’ Chemistry Institute of Kazan Federal University, 18 Kremlevskaya Street, 420008 Kazan, Russia; leongard87@mail.ru

**Keywords:** cyclophanes, resorcin[4]arenes, calix[n]arenes, thiacalix[n]arenes, pillar[n]arenes, self-assembly, silver nanoparticles

## Abstract

Silver nanoparticles (AgNPs) are an attractive alternative to plasmonic gold nanoparticles. The relative cheapness and redox stability determine the growing interest of researchers in obtaining selective plasmonic and electrochemical (bio)sensors based on silver nanoparticles. The controlled synthesis of metal nanoparticles of a defined morphology is a nontrivial task, important for such fields as biochemistry, catalysis, biosensors and microelectronics. Cyclophanes are well known for their great receptor properties and are of particular interest in the creation of metal nanoparticles due to a variety of cyclophane 3D structures and unique redox abilities. Silver ion-based supramolecular assemblies are attractive due to the possibility of reduction by “soft” reducing agents as well as being accessible precursors for silver nanoparticles of predefined morphology, which are promising for implementation in plasmonic sensors. For this purpose, the chemistry of cyclophanes offers a whole arsenal of approaches: exocyclic ion coordination, association, stabilization of the growth centers of metal nanoparticles, as well as in reduction of silver ions. Thus, this review presents the recent advances in the synthesis and stabilization of Ag (0) nanoparticles based on self-assembly of associates with Ag (I) ions with the participation of bulk platforms of cyclophanes (resorcin[4]arenes, (thia)calix[n]arenes, pillar[n]arenes).

## 1. Introduction

Silver nanoparticles (AgNPs) are attractive nanomaterials, especially promising in such fields as colorimetric sensors, bactericidal materials and components of electrochemical sensors [1,2,3]. In contrast to gold nanoparticles, silver nanoparticles can be readily synthesized from relatively stable silver (I) salts using a wide range of reducing agents [4]. Moreover, silver nanoparticles find application as sterilizing nanomaterials in medicine. A unique feature of AgNPs is their slow release of silver (I) ions, which allows using them as bactericidal agents [5,6,7]. Silver nanoparticles are promising catalytically active nanomaterials, which can enhance some reactions for dye and drug synthesis [8,9].

These applications require surface modification of AgNPs that would allow to adjust their association with the target substrate or to offer higher selectivity towards association with bacterial cell walls. Depending on the application, AgNPs must be modified with different functionality. For electrochemical sensors and antibacterial nanomaterials, it is necessary to introduce functionality, providing AgNPs with selectivity of association [10]. However, the strategy of creating AgNP-based colorimetric sensors is different [11]. In order to obtain a wavelength shift in the presence of target a substrate, it is necessary to introduce functionalities that would not only bind to the target substrate but also induce aggregation of AgNPs [12].

One of the most promising strategies for creating selective nanoparticles is using supramolecular agents capable of forming host–guest complexes with the target substrate [13]. Cyclophanes are promising macrocyclic 3D host molecules, which are versatile building blocks for self-assembled materials [14,15,16]. Cyclophanes have been reported as selective host molecules for recognition of inorganic ions, and organic molecules and have been shown promising in biopolymer binding [17,18,19,20,21,22]. The vast opportunities for obtaining various functional derivatives and rich spatial geometry of cyclophanes offers the means necessary for achieving this goal [23].

This review summarizes the modern achievements in design and application of AgNPs functionalized with cyclophanes [24,25,26,27,28,29,30,31,32,33,34,35,36,37,38,39,40,41,42,43,44,45,46,47,48,49,50,51,52,53,54,55,56,57,58,59,60,61,62,63,64,65,66,67,68,69,70,71,72,73,74,75,76,77,78,79,80,81,82,83,84,85,86,87,88,89,90,91,92,93,94,95,96,97,98,99,100,101,102,103,104,105,106,107,108,109,110,111,112,113,114,115,116,117,118,119,120,121,122,123,124,125,126,127,128,129]. Several classes of cyclophanes were chosen: calix[n]arenes, their sulfur bridged analogue thiacalix[4]arenes, resorcin[4]arenes and pillar[n]arenes (Figure 1). Other cyclophanes are out of current review because such publications are scarce. While in this review we report recent findings in the field of cyclophane-functionalized AgNPs synthesis and application (Figure 2), our review has several key features. Specific attention is paid to comparing unique substrate recognition patterns that are inherent for each cyclophane type. Also, it is widely known that cyclophanes may form nanosized supramolecular associates with silver ions, and, on the other hand, there are a number of reports on the synthesis of silver nanoparticles in the presence of cyclophanes. While both are well known to the scientific community, to the best of our knowledge there have been no studies in which possible supramolecular self-assembly of cyclophane with silver ions prior to reduction would be investigated.

Using three-dimensional macrocyclic stabilizing agents at the stage of AgNPs synthesis provides unique means to adjust the sizes of nanoparticles at the growth stage, which is promising in the nanomaterial design field. Functionalization of AgNPs with cyclophanes allows achieving high selectivity towards various inorganic, organic and biomacromolecular substrates. It opens wide opportunities in developing novel selective antibacterial agents, surface-enhanced Raman scattering (SERS)-based colorimetric sensors and electrochemical sensors. The review presents achievements in cyclophane based AgNPs, which are especially promising for nanomedicine and diagnostics.

## 2. Supramolecular Self-Assembly of Calix[n]arenes with Ag(I) Ions and Calix[n]arene-Based AgNPs

Calix[n]arenes are the most wide-known cyclophane platforms. Most of the synthetic routes, patterns of substrate recognition and supramolecular self-assembly were realized on this platform for the first time and afterwards were successfully adapted to other cyclophanes. Ease of modification of lower and upper rims of calixarenes have made them versatile building blocks for supramolecular materials and surface modifiers for adjusting receptor and self-assembly properties of colloidal materials. Separate modification of upper and lower rims of calixarenes has become one of the common practices to obtain amphiphilic host molecules for Langmuir–Blodgett films and consequently for modification of nanoparticles, allowing to alter their lipophilicity [24].

### 2.1. Self-Assembly of Calix[n]arenes with Ag(I) Ions

Studying the self-assembly of calixarene derivatives with silver ions can help in understanding the role of calix[n]arene in the synthesis of AgNPs. Usually the role of calixarene is described as a stabilizer (Table 1). It is worth noting that most calixarene derivatives are poorly soluble in water, which complicates their use in the synthesis of AgNPs via the most common route (reduction with sodium borohydride). One approach to solve this problem is the creation of amphiphilic structures based on this cyclophane.

Tetrasubstituted amphiphilic calix[4]arene **CA-1** (Figure 3) containing alanine fragments on the upper rim and decyl groups on the lower rim was synthesized [25]. It was shown that macrocycle **CA-1** can self-assemble into various structures depending on the solvent pH. It forms a spherical structure at pH = 3 and a hollow, necklace-like structure of 500 nm diameter at pH 7. It has been shown that addition of silver ions leads to a three-dimensional dendrite nanostructure. The authors supposed that calix[4]arene **CA-1** acted as a stabilizer and shape controller in the diffusion-limited aggregation process. The obtained results can be used in the design of microelectronic devices or nanometer-scale electrodes.

Double rosette assemblies of calix[4]arenes **CA-2** and **CA-3** with barbituric acid or cyanuric acid derivatives and silver were shown to reduce by electron beam at 200 keV *in situ* in conditions of a transmission electron microscopy (TEM) experiment [26]. Such method of reduction allows to obtain small monodisperse nanosized (d = 2 nm) AgNPs. The obtained results prove that association with cyclophanes allows to synthesize very small AgNPs.

Self-assembly of calix[6]arene containing imidazole groups on the lower rim and three sulfonato groups on the upper rim was studied [27]. It was shown that at concentrations of 10^−4^ M and higher, macrocycles formed multilamelar vesicles (50–250 nm diameter). Addition of silver (I) ions to the system leads to formation of nanomicelles with size of 2.5 nm. The authors suggest that the obtained micelles can be used in hosting of a guest, either in the calixarene core itself or in the heart of the assemblies.

Calixarenes upon self-assembly with silver ions can form not only micellar structures but highly regular networks of coordination polymers. It is caused by strict fixation of coordinating fragments relative to the macrocyclic rim. Possibility of functionalization of a macrocyclic platform with various functional groups, spacers and various conformations available for calixarenes has made them a versatile block for metal–organic frameworks with silver ions. For example, three component coordination polymers of silver ions with *p-*sulfonatocalix[4]arene **CA-4** and ethylenediamine were reported [28]. Similarly, carboxyl derivatives of calix[4]arene **CA-5** and **CA-6** [29], calix[4]arenes **CA-7** with allyl groups [30] and **CA-8** with pyridyl fragments [31] were shown to form regular crystalline structures. The last-mentioned material has been successfully used for recognition of nitrobenzene and photodegradation in aqueous solutions. There is an example of macrocycle **CA-5** and **CA-9** self-assembly in the presence of silver ions leading to the formation of dimeric capsules, which over time crystallize into three-dimensional molecular architecture [32].

The publications reporting self-assembled supramolecular nanostructures of calixarenes with silver ions are still scarce; however, considering the long period over which calixarenes have been known to the scientific community and intensely studied, it leads us to an indirect conclusion: such supramolecular association patterns are most likely uncommon for calixarenes. Molecular design approaches for creating supramolecular nanoparticles instead of regular three-dimensional MOFs (metal–organic frameworks) are unknown.

### 2.2. Synthesis and Application of Calix[n]arene-Based AgNPs

Calix[n]arene-based AgNPs have been synthesized using various methods [15,33]. The most commonly used approaches are chemical, electrochemical and photochemical reduction. Historically, the simplest and most studied method is chemical reduction (Table 1). Sodium borohydride is the most popular reducing agent (along with sodium citrate, hydrazine hydrate and molecular hydrogen). However, the choice of functional groups of substituents is a nontrivial task in the design of calixarene-based AgNPs. Basically, functional groups should allow association with AgNP surfaces. Especially in the case of calixarenes, the nature and geometry of functional groups (which are usually strictly preorganized by macrocyclic platform) dictate possible interaction with the target substrate. Therefore, this factor sufficiently limits all the possible substrate recognition patterns achieved for calixarenes, restricting possible functional groups to only those capable of interacting with AgNP surfaces. It could be overcome by introducing functional groups of different nature, but in practice (as we can see in articles listed below), mostly calixarenes modified with functional groups capable of association with AgNPs were reported.

#### 2.2.1. Calix[n]arene-Based AgNPs Obtained Using Chemical Reduction

In supramolecular chemistry and in the chemistry of macrocyclic compounds, fragments of sulfonic acids usually are implemented to increase solubility in polar media (in water), giving surficial active properties and self-assembly based on acid–base or ionic interactions. In addition to mentioned directions, sulfonic acids are capable of coordinating some ions (e.g., silver ions) and can coordinate to the surface of metallic silver (0) nanoparticles. Sulfonic acid derivatives of calixarene are capable not only to associate with silver ions but also with nanoparticles: silver nanoparticles functionalized with sulfonic acid derivatives of calix[n]arenes were reported.

Significant contribution in the synthesis and application of sulfonatocalix[n]arene-stabilized AgNPs was made by Prof. A. W. Coleman’s group [33]. This group synthesized a number of sulfonic acid calixarene derivatives **CA-4** and **CA-10**–**CA-17** (Figure 4), which were used for surface modification of silver nanoparticles. Possibility of interacting sulfonatocalix[n]arene-based AgNPs with cationic, neutral and anionic surfactants was studied. It was shown that only *p-*sulfonatocalix[4]arene derivatives **CA-4** and **CA-11** interacted with cationic surfactants [34]. Toxicity of the nanoparticles obtained towards model pheochromocytoma neuronal-like cells was studied, and it was shown that synthesized nanoparticles were nontoxic up to 100 mg/mL concentration. It was shown that treatment of neural pheochromocytoma cells with such nanoparticles with calix[n]arenes leads to lowering of reactive oxygen species [35].

It was shown that AgNPs capped with nine different sulphonated calix[n]arenes (**CA-4** and **CA-10**–**CA-17**) were tested for their antibacterial effects against *Bacillus subtilis* and *Escherichia coli* at concentration of 100 nM in calix[n]arene [36]. It was determined that antibacterial properties depend on the size of the macrocycle ring. It was shown that calixarenes (**CA-11**, **CA-14** and **CA-17**) with sulfonate groups on the lower and upper rims were active against Gram-negative bacteria. Derivatives of calix[6]arene **CA-13** containing sulfonate groups only on the lower rim has pronounced bactericidal activity.

It was reported [37] that interaction of *p-*sulfonatocalix[4]arene **CA-4** based AgNPs with nucleotides, nucleosides and desoxynucleosides leads to selective complexation, which is expressed by color change due to aggregate formation. There is a clear difference for purine and pyrimidine: for pyrimidine nucleotides, a new absorption band at 550 nm appears, and a color change from yellow to orange, red or pink is observed; for purine nucleotides, an absorption band is observed at 580–590 nm.

Molecular recognition of cytosine with *p-*sulphonatocalix[4]arene **CA-4** in solution, in the solid-state and on the surface of hybrid silver nanoparticles was studied [38]. It was found that the ratio of **CA-4**:cytosine changed depending on the state (from 1:1 in solution to 1:4 in solid phase). It was demonstrated that cytosine initiated aggregation of **CA-4**-based AgNPs.

Silver nanoparticles stabilized with oleic acid in organic solvents were obtained [39]. Adding of *p*-sulfonatocalix[4]arene **CA-4** to this system leads to formation of an inclusion complex of oleic acid and **CA-4**, which allows to transfer silver nanoparticles from the organic phase into the aqueous phase.

AgNPs modified with *p-*sulfonatocalix[4]arene **CA-4** were obtained by adding sodium borohydride to a mixture of **CA-4** and silver nitrate at room temperature for 5 min [40]. AgNP size was determined by TEM and was 8.0 ± 1.0 nm. A colorimetric sensor was made based on these AgNPs for detecting histidine in water. High selectivity of this sensor allows to quantitatively determine histidine concentrations down to 5 × 10^−6^ M.

Presence of additional binding sites also opens new opportunities towards forming cascade composite structures based on inorganic materials. *p-*Sulfonatocalix[4]arene **CA-4**-modified AgNPs were synthesized by interaction of AgNO_3_–calixarene mixture with NaBH_4_ [41]. Solutions of nanoparticles obtained upon addition of metal ions at pH 10 have different colors. The authors explain such color diversity by the assembly size of **CA-4** AgNPs on the metal hydroxide crystals, which depend on the added metal species, leading to a different localized surface plasmon resonance band. Therefore, **CA-4**-AgNPs also may be used for discerning different metals by their color.

Highly stable AgNPs functionalized with *p-*sulfonatocalix[n]arenes **CA-4** (*n* = 4) and **CA-15** (*n* = 8) were obtained [42]. Interaction of the synthesized AgNPs with a number of pesticides (iprodione, pyrimethanil, thiabendazole, optunal, parathion-methyl, methomyl and acetamiprid) was investigated. It was shown that the AgNPs with calix[4]arene **CA-4** can be implemented as colorimetric probes to quantitatively determine optunal at concentrations down to 1 × 10^−7^ M.

Presence of a large flexible hydrophobic cavity in sulfonatocalix[6]arene **CA-12** opens the possibility of complexation with fluorescent ligands. Such double functionality opens the possibility of making three-component systems in which a fluorescent ligand acts as a bridge between cyclophane-modified silver nanoparticles [43]. A highly sensitive H_2_PO_4_^−^ colorimetric sensor was developed based on self-assembly of **CA-12**-AgNPs with a dipyrene derivative. The authors report that this colorimetric sensor can also realize real-time quantitative detection for H_2_PO_4_^−^ with a detection limit of 1 × 10^−7^ M.

Similarly, calixarene derivatives can be used for creating a graphene–AgNPs nanocomposite in a mixture of silver nitrate, graphene oxide and potassium hydroxide. Ag–graphene nanocomposites were synthesized by utilizing a continuous hydrothermal flow synthesis (in water overheated up to 450 °C, 24.1 MPa) process using sulfocalix[6]arene **CA-12** as an effective particle stabilizer [44]. The authors report low cytotoxicity and high activity against *Escherichia coli* (Gram-negative) and *Staphylococcus aureus* (Gram-positive) bacteria.

Modification of AgNPs with sulfonic acid derivatives of calixarenes **CA-4**, **CA-12** and **CA-15** also allows stabilizing enzymes [45]. It was shown that *p-*sulphonatocalix[8]arene **CA-15**-stabilized AgNPs can increase efficiency and stability of Saburopin (serpin enzyme from *Eubacterium saburreum* possessing proteolytic activities). The authors report that in the presence of **CA-15**–AgNPs, enzymes have a high inhibition effect in the pH range of 2–10 and in the temperature range of 10 to 40 °C, while remaining active even at 70 °C.

AgNPs were obtained using a series of anionic calix[4]arenes with sulfonate (**CA-4**, **CA-10** and **CA-11**), carboxylate (**CA-18** and **CA-19**) and phosphonate (**CA-20**) groups [46]. Selectivity of obtained nanoparticles was studied on a series of three active pharmaceutical ingredients (API): chlorhexidine (antiseptic), chloramphenicol (neutral antibiotic) and gentamycin sulfate (aminoglycosidic antibiotic). It was shown that affinity to APIs studied depended on the nature of calix[4]arene and API.

The binding of AgNPs modified with calix[4]arenes **CA-4** and **CA-20** with regard to various serum albumins (bovine, human, porcine and sheep serum albumins) has been studied by variable temperature fluorescence spectroscopy [47]. The studied albumins have different fluorescence quenching effects according to the nature of the anionic calixarene. The results obtained by the authors may help to discriminate among the different species.

AgNPs with diameter about 2 nm were obtained using *p-*phosphonated calix[n]arenes **CA-21** (*n* = 4), **CA-22** (*n* = 5), **CA-23** (*n* = 6) and **CA-24** (*n* = 8) as templates and surfactants for stabilization [48]. Hydrogen was used as a reducing agent. The authors show that the key factor affecting particle sizes and speed of reaction was the solution pH. The smallest monodisperse particles were obtained at pH = 12.

Due to preorganized hydrophobic cavity, one promising application of AgNPs functionalized with cyclophanes is their use for recognition of polycyclic aromatic hydrocarbons [49]. Calix[4]arenes (**CA-25**–**CA-27**)-modified AgNPs were investigated for affinity towards a series of polycyclic aromatic hydrocarbons (PAHs) (pyrene, triphenylene, benzo[*c*]phenantrene, anthracene, coronene, chrysene, dibenzoanthracene and rubicene) [50]. AgNPs modified with disubstituted derivatives of calix[4]arene **CA-26** with two ester groups have selectivity towards polycyclic aromatic hydrocarbon guest systems bearing four benzene rings, mainly pyrene. It opens the opportunity for using functionalized calix[4]arenes in the design of highly sensitive and selective sensors of PAHs.

Dithiocarbamate calix[4]arene **CA-28** functionalized AgNPs were shown to detect traces of aromatic hydrocarbons using the surface-enhanced Raman scattering (SERS) method [51,52,53]. The effectiveness of this system was checked for a group of PAHs: pyrene, benzo[c]phenanthrene, triphenylene and coronene. High selectivity towards pyrene, benzo[c]phenanthrene and triphenylene containing four aromatic rings was demonstrated. For these compounds, affinity constants have close values, and detection limits are quite similar (10^−7^, 10^−8^ M). In the case of coronene, the sensitivity was higher, which was explained by formation of a complex in which cavities of two calixarene molecules participated, leading to aggregation of nanoparticles.

Also, there are a series of articles reporting unsubstituted *p-tert*-butylcalix[n]arenes as AgNP stabilizers. A unique approach to the synthesis of AgNPs was developed [54] in a preassembled three-component Ag–ethylenediamine *p-tert-*butylcalix[4]arene **CA-25** coordination complex. Subsequent removal of amine under heating leads to reduction of Ag^+^ ions and formation of Ag nanocrystals (d = 9.4 nm) with 29 Ag atoms in a single crystal.

AgNPs based on calix[6]arene **CA-12** with diameters of about 7 nm are capable of interacting with sanguinarine (antitumor, antibacterial, antioxidant and anti-inflammatory drugs) [55]. The nanoparticle–sanguinarine complex has shown to lower toxicity towards two cell lines (normal CHO and cancerous A549).

In the research group of Prof. D. Tian [56], AgNPs modified with the tetracarboxylate derivative of *p-tert-*butylcalix[4]arene **CA-29** in the 1,3-alternate conformation containing triazole spacers were obtained. A highly sensitive colorimetric sensor for Cu^2+^ was developed with the detection limit of 2.5 × 10^−6^ M. Moreover, the authors note that increasing Cu^2+^ concentration leads to increasing sizes of AgNPs and forming larger three-dimensional aggregates.

AgNPs modified by sulfonatocalixarene **CA-30** with thiol groups on the lower rim were prepared [57]. Obtained AgNPs were used for detecting trace concentrations of Fe^3+^ ions in aqueous medium. The sensor obtained has several advantages compared to reported colorimetric sensors: fast response time (<80 s) to ferric ions and long shelf-life (>4 weeks). The biosensor has been successfully applied to estimate ferric ions in human blood serum as well as in human hemoglobin.

Using thiol derivatives of cyclophanes for surface modification of metal nanoparticles can be considered as a more straightforward approach. AgNPs (d = 3 ± 1 nm) modified with disubstituted derivatives of calix[4]arene **CA-31** functionalized by thiol groups were obtained [58]. These nanoparticles were shown to be hosts for gold nanoparticles coated with 1-dodecanethiol and *N*-alkylpyridinium fragments. Specific recognition of pyridinium fragments of gold nanoparticles by **CA-31**-modified AgNPs was demonstrated by UV spectroscopy and dynamic light scattering (DLS) methods in dispersions.

While the abovementioned examples were limited to calixarenes functionalized with alkyl groups and fragments capable of interacting with AgNP surfaces, there are few examples of introducing functional fragments specific to target substrate recognition. Using chiral carboxylic acid fragments allows chiral substrate recognition while maintaining association to AgNPs surface. AgNPs based on chiral calix[4]arene **CA-32** containing R-mandelic acid fragments were obtained [59]. These AgNPs with diameters about 9 nm can differentiate N-Fmoc-d/l-aspartic acid (D-FAA and L-FAA), with higher sensitivity of chiral calixarene-modified AgNPs towards L-FAA. Several detection methods were compared, and DLS has been shown to be more sensitive to nanoparticle aggregation than UV spectroscopy (500-fold higher sensitivity of DLS method compared to UV spectroscopy).

Historically, calixarene is one of the first macrocyclic platforms that were used as a stabilizer in the synthesis of AgNPs. Most often, derivatives of calix[n]arenes containing charged sulfonate, carboxylate and phosphonate groups are used for their functionalization. Keeping in mind that, especially for calixarenes, the nature of functional groups dictates selectivity, it definitely limits their potential as host molecules. The obtained calixarene-based AgNPs are used to design and create drug delivery systems, colorimetric sensors, antibacterial agents and separation and selective recognition of various biologically important substrates.

#### 2.2.2. Calix[n]arene-Based AgNPs Obtained Using Photochemical Reduction

The photochemical synthesis method offers a reasonable potential for the synthesis of shape- and size-controlled calixarene-based AgNPs. It is also worth noting that the photochemical reduction method is environmentally friendly; therefore, this method can be attributed to “green chemistry”. Excluding reducing agents from the reaction mixture simplifies the reduction of calixarene-silver ions, allowing to expect formation of AgNPs in a more associate–controlled fashion (Table 1).

Calixarene **CA-33** functionalized with two pyridyl groups linked to cyclophane with a triazole spacer [60] was found to selectively bind Fe^3+^ (1:1 stoichiometry ratio, K = 5.35 × 10^−4^ M^−1^). Calixarene **CA-33** was used for synthesis of **CA-33**-AgNPs (diameter 10 ± 1 nm) by reduction of AgNO_3_ in methanol using ultraviolet radiation (*λ*_max_ = 365 nm). A colorimetric sensor for Fe^3+^ was designed based on **CA-33** AgNPs. The authors report strong aggregation of **CA-33** AgNPs at higher concentrations of Fe^3+^.

AgNPs (d~5 nm) were obtained using disubstituted calix[4]arene **CA-34** with hydrazide and thiophene fragments [61]. These nanoparticles were prepared by mixing a calixarene solution in methanol with aqueous solution of AgNO_3_ in the ratio of AgNO_3_/**CA-34** 1:15. The obtained reaction mixture was treated with sunlight. Temperature effects were studied at 20, 50 and 80 °C. It was shown that nanoparticle sizes did not depend on temperature significantly. Obtained nanoparticles can selectively recognize Hg^2+^ and Hg^0^ in solution and vapor phases, respectively, with distinct color change. The detection limits reported for Hg^2+^ by UV–Vis and amperometry are 0.5 nM (0.1 ppb) and 10 nM (2 ppb), respectively.

AgNPs with diameters less than 5 nm were obtained by photochemical reduction of silver nitrate (1 mM) in water in the presence of *p-*phosphorylated calix[n]arenes **CA-21**, **CA-23**, **CA-24** and *O*-alkyl derivatives of **CA-21** (**CA-35** and **CA-36**) (0.25 mM) [62]. Photochemical reduction is one of the key methods of green chemistry. The experiment was conducted using a 100 W UV lamp emitting 365 nm light. At pH = 9, the size of obtained nanoparticles was 3.6 ± 1.2 nm.

Alkylthiol derivatives of calix[8]arene **CA-37** were used for modification of monometallic and bimetallic nanoparticles [63]. AgNPs were obtained by reducing Ag^+^ in the presence of calix[8]arene **CA-37** in ethanol. In the results, small spherical nanoparticles were obtained with size dependence on silver salt concentration and counter ion. The structure of the calix[8]arenes and their anchoring on the AgNPs surface cause some accessibility to the surface of the AgNPs, which is very important for applications in catalysis. Potential catalytic applications of the obtained **CA-37** AgNPs were studied using the reduction of 4-nitrothiophenol as a model reaction.

It can be concluded that very small (5 nm or less) calixarene-based AgNPs can be obtained using the photochemical reduction method, which are smaller compared to AgNPs obtained by common chemical reduction route. Unfortunately, shapes and sizes of associates of calixarene with silver ions prior to reduction were not studied, while it could shed light on the effect of supramolecular association on size and morphology of the AgNPs formed. These results make this method very promising.

#### 2.2.3. Calix[n]arene-Based AgNPs Obtained Using Electrochemical Reduction

The use of the electrochemical method for synthesis of AgNPs is one of the promising areas due to a number of advantages. Electrochemical methods make it possible to obtain nanoparticles with high purity using simple techniques, while controlling the particle size can be done by adjusting the current density or the applied potential. It is also worth noting that this method is environmentally friendly, since toxic chemical reducing agents are not used. On the other hand, the electrochemical reduction proceeds in a heterogeneous environment (and with participation of AgNPs, resulting in their clustering), which leads to certain limitations of the method for size-control of the nanoparticles formed. Usually this method is implemented for constructing hybrid nanostructured electrochemical sensors, and introduction of AgNPs allows to increase sensitivity of electrodes.

Novel electrochemical sensors based on *p*-sulfonatocalix[6]arene **CA-12**-modified AgNPs coated glassy carbon electrode for methyl parathion as a model of nitroaromatic organophosphates were developed [64]. The obtained sensors can determine methyl parathion in the concentration range of 1 × 10^−8^ to 8 × 10^−5^ M with lower detection limit of 4.0 × 10^−9^ M. The authors report that reproducibility and stability of the sensor obtained was higher than those for enzyme-based electrodes.

AgNPs with diameters in a range of 40–70 nm were obtained by an electrochemical method using *p-tert*-butylcalix[4]arene **CA-25** on glassy carbon as the matrix [65]. The authors report that nifedipine (calcium channel blocker) can be detected in concentration range (0.8–60 μM) with detection limit at 0.72 μM. This electrode has a number of attractive properties: high stability, reproducibility of signal, high sensitivity, quick response and low detection limit.

AgNPs were synthesized on glassy carbon electrodes coated with *p-tert*-butylcalix[4]arene **CA-25** and *p-tert*-butylcalix[6]arene **CA-38** [66]. The authors demonstrate that the presence of calixarene on the electrode’s surface allows to control nanoparticle sizes and to prevent agglomeration. Cyclic voltammetry has shown that AgNPs on glassy carbon electrode have pronounced catalytic activity to reduce flutamide, which is a widely used nonsteroidal anti-androgen drug in prostate cancer treatment. The modified electrode shows linear signal response in differential impulse voltammetry in a range 10–1000 μM with detection limit at 9.33 μM for flutamide.

An approach to synthesize AgNPs by electrochemical reduction of Ag^+^ [67] on the glassy carbon electrode, modified with *p*-isopropylcalix[6]arene **CA-39** was reported. Obtained AgNPs were able to efficiently catalyze reduction of hydrogen peroxide.

An *in situ* synthesis route was reported [68] for preparation of a layer of AgNPs on the silicon surface using an immobilized supramolecular layer as silver ion reducing agent. First, *p-*methoxycalix[7]arene **CA-40** was covalently immobilized on the silicon surface, and 4-methoxyphenol fragments were demethylated to hydroquinone fragments. Then, AgNPs were immobilized into this layer by reducing with Na_2_S_2_O_4_. AgNPs had plate-like morphology with a diameter of 10 nm and height of 2 nm. Repeating redox cycles of calix[7]hydroquinone **CA-41** can be implemented to vary sizes of plates in a range of 100 nm and more. Further study of AgNP properties in the same research group [69] was conducted, and it was shown that nanoparticles obtained were capable of photodegradation of various organic dyes (methyl orange, methylene blue and rhodamine chloride).

Thus, calixarene-based AgNPs obtained using the electrochemical method are mainly used as part of electrochemical sensors to determine biologically important substrates, and they can also be used as catalysts and components for photodegradation of dyes. Usually, electrochemical reduction leads to rather large sizes of AgNPs when compared to photoreduction and chemical reduction. On the other hand, the electrochemical method has definitely allowed to overcome several drawbacks of these methods: it allows to choose other groups for calixarene functionalization and using hydrophobic calixarenes.

## 3. Supramolecular Self-Assembly of Thiacalix[4]arenes with Ag(I) Ions and Thiacalix[4]arene-Based AgNPs

Among cyclophanes mentioned, thiacalixarenes are characterized by high conformational flexibility and larger sizes of macrocycles compared to calixarenes [70]. The unique feature of thiacalix[4]arenes is the presence of sulfur bridge atoms, which are capable of coordinating with transition metal ions, having high affinity to silver ions. For the most widespread representative of thiacalixarene-based synthetic platforms, *p-tert-*butylthiacalix[4]arene, three conformations are readily available (cone, partial cone and 1,3-alternate). This opens a unique possibility of distributing substituents on both sides from the macrocyclic rim, which in combination with introducing various substituents allows to reach various substrate selectivities. Even more interesting is the applicability of this feature in various branches of material science and supramolecular chemistry. It allows to achieve unique association patterns, adjust polymer properties and vary aggregation of nanoparticles [71]. Due to the unique geometry of thiacalixarenes, they tend to form nanocage associates [72]. Therefore, in regard to silver ions and nanoparticles, thiacalixarenes are unique, allowing new patterns of self-assembly and nanoparticle functionalization.

### 3.1. Self-Assembly of Thiacalix[4]arenes with Ag(I) Ions

High affinity of silver ions and the presence of four sulfur bridge atoms, capable of coordinating silver ions, are among the key properties that have attracted the attention of the research community to thiacalixarenes. In the scientific group of Prof. M. W. Hosseini, molecular tectonics based on the formation of silver coordination networks by thiacalix[4]arene derivatives were investigated. The geometry of the final self-assembly was determined by both structural and coordination features of the organic tecton. The authors suggested that the obtained perspective molecular architectures can be used in catalysis, optics, electronics and magnetism [73,74,75,76,77,78,79,80,81,82].

While there are many reports on MOFs of silver-coordinated thiacalixarenes, it should be noted that additional coordinating sites (sulfur bridge atoms) can play dramatic roles in association with silver ions. For example, in contrast to any other cyclophanes [83], *p-tert-*butylthiacalix[4]arene **TCA-1** (Figure 5) can form a metal organic framework, where silver ions are coordinated by bridge groups. Therefore, when functionalized with additional groups capable of coordinating silver ions, these cyclophanes can have unique association behavior with silver ions (and other transition metal ions). Indeed, for thiacalixarenes, significant parts of publications are related to forming submicron and nanosized supramolecular associates in the presence of silver salts.

Thiacalixarene’s ability of coordinating silver ions has opened the perfect opportunity for using them in electrochemical sensors for silver ion detection [84]. Associates of thiacalixarene derivatives **TCA-2** and **TCA-3** in cone, partial cone and 1,3-alternate conformations with silver ions form solid antimicrobial coatings [85].

Functionalization of thiacalixarene with propylsulfonate fragments (**TCA-4**, cone) allows to realize supramolecular self-assembly into unique shaped associates [86]. Sulfonate substituents are known as ligands for silver ions and, therefore, can provide additional coordination sites for silver ions, which is possibly the main reason of forming fractal associates. While self-assembly of propylsulfonate-functionalized (thia)calixarenes leads to formation of polydisperse submicron-sized micelles, in the presence of silver ions (equimolar ratio) polydispersity significantly decreases. In contrast, for calix[4]arene with similar functional fragments, the hydrodynamic diameter of the particles is 211.1 ± 11.6 nm (polydispersity index = 0.41 ± 0.01), and for the thiacalix[4]arene **TCA-4** it is 95 ± 7 nm (polydispersity index = 0.23 ± 0.01). The presence of additional coordinating sites greatly changes not only the self-associate sizes and stability but also their behavior: fractal associates are formed.

In the case of thiacalixarene three conformational isomers are readily accessible (cone, partial cone and 1,3-alternate). In some cases, conformation can play an interesting role in shape and morphology of associates with Ag^+^. Alkylamide derivatives of thiacalixarene **TCA-2**, **TCA-3** and **TCA-5**–**TCA-10** were found to form associates with silver ions with different sizes and shapes as determined by atomic force microscopy (AFM), when interacting with silver ions. Most flat aggregates are formed in the case of 1,3-alternate stereoisomers (2 nm height), while the height of cone associates is 27 nm [87]. The associate sizes are in a range of 62.5–491.6 nm (determined by AFM), while the morphology of aggregates is more dependent on conformation of thiacalixarene derivatives: disk-shaped for partial cone and 1,3-alternate conformations and spherical or elongated structures for cone conformation are formed.

High affinity to silver ions caused by bridging sulfur atoms allows introducing additional binding sites by lower rim functionalization while retaining association with silver ions. This allows not only to adjust morphology of supramolecular associates, but it also opens opportunity to use these associates as receptors. It was reported that associates of benzotriazole and arylhydrazide thiacalixarene derivatives with silver ions can interact with dicarboxylic acid, resulting in strong association dependency on the structure of dicarboxylic acid. It was reported that the sizes and morphology of arylhydrazide and benzotriazole derivatives of thiacalixarene **TCA-11–TCA-16** commutatively depend on the sequence of adding silver ions and dicarboxylic acid, and the size of three-component supramolecular associates significantly depends on the structure of dicarboxylic acids [88].

Functionalization of thiacalixarene with carboxylic, hydrazide and amino-groups along with 12 guanidine fragments (**TCA-17**–**TCA-19**) even more significantly enhances coordination with carboxylic acids [89]. Spherical associates of thiacalixarenes with silver ions change morphology upon addition of oxalic acid to granular aggregates.

Assembly of zwitterion-functionalized thiacalixarenes **TCA-20** and **TCA-21** with silver ions not only increased the hydrophilicity of resulting associates, but it also demonstrated significant dependence of supramolecular associate sizes on the length of the spacer connecting the quaternary ammonium fragment to the sulfonate group (197 and 332 nm for propanesulfonic **TCA-20** and butanesulfonic **TCA-21** derivatives) [90].

Adding positively charged fragments to thiacalixarene structures **TCA-22**–**TCA-26** opens the route to creating three-component associates with dye molecule (fluorescein), which allows fluorimetric detection of bovine serum albumin (BSA). Sizes of supramolecular associates with silver ions and fluorescein in the range of 10–67 nm are comparable with sizes of associates with silver ions only (8–88 nm). Among cyclophanes studied, three-component systems based on methyl-ammonium **TCA-22** and phthalimidopropyl-ammonium **TCA-23** derivatives have the highest affinity to bovine serum albumin [91].

Thus, the presence of bridging sulfur atoms, as well as functional groups of substituents, is a fundamental factor in the design and creation of supramolecular systems based on silver cations and derivatives of thiacalix[4]arene. Among functional groups, amide, hydrazide and pyridinium groups can be distinguished. The choice of such substituents is explained from the point of view of the Pearson acid–base concept: the “soft” silver cation more effectively interacts with the “soft” bridging sulfur atom, as well as substitute fragments, and several variants of the coordination of silver ions with thiacalix[4]arene molecules can be suggested (Figure 6). The obtained systems can be used for targeted drug delivery and selective recognition of proteins, amino acids and hydroxyl acids.

### 3.2. Synthesis and Application of Thiacalix[4]arene-Based AgNPs

While there is a number of publications reporting self-assembly of thiacalixarenes with silver ions, unfortunately there are only several reports on reduction of self-assembled structures to AgNPs (Table 1).

It was shown that self-assembly of **TCA-3** and **TCA-8** with AgNO_3_ under laser irradiation (633 nm) in DMF led to formation of thiacalixarene-coated silver nanoparticles [92]. It was shown that temperature (varied in a range of 2–50 °C) had significant impact on the size of nanoparticle aggregates: average size increased with increasing temperature. The largest associates of AgNPs were observed for **TCA-3** in partial cone conformation.

A hydrazide derivative of thiacalixarene **TCA-5** was shown to reduce silver ions in the work of Prof. V. K. Jain’s research group [93]. Relatively uniform 20 nm spherical AgNPs were formed over a pH range of 5–9. The interaction behavior of **TCA-5** AgNPs with different amino acids was investigated using spectrophotometry and spectrofluorimetry. Among the amino acids tested, only tryptophan and histidine showed fluorescence quenching and fluorescence enhancement, respectively. **TCA-5** AgNPs were reported to effectively reduce the levels of Gram-positive bacteria, Gram-negative bacteria and fungi.

Due to the unique affinity to silver ions, the tendency of thiacalixarenes to form nanocages is especially promising as protection for silver nanoclusters. Nonfunctionalized *p-tert*-butylthiacalix[4]arene **TCA-1** has been shown as an effective protecting agent for silver nanoclusters (containing 35 and 34 Ag atoms) [94].

Dopamine is a known reducing agent of silver ions. It was shown that the tetradopamide derivative of thiacalixarene, **TCA-27,** can chemically reduce silver ions. There are certain similarities with reports of Prof. V. K. Jain’s research group: reduction proceeds slowly starting with the formation of associates with silver ions with their further reduction to form 4–6 nm particles. Chemical reduction takes several hours, and the obtained coating is more uniform and acts as more effective electrochemical sensing nanomaterial when compared to electrochemically reduced silver salt on the **TCA-27**-coated electrode. With variations in other components of electrochemical sensors, these nanoparticles were successfully used for dopamine [95] and ochratoxin determination [96]. This approach has allowed to obtain electrochemical sensors for cholinesterase, where quaternary ammonium derivatives of thiacalixarene **TCA-22–TCA-24** act as cholinesterase inhibitors [97]. The same nanostructures were implemented for impedimetric detection of DNA damage in sensors containing neutral red [98].

Electrochemical reduction of silver ions on electrodes coated with thiacalixarene-functionalized oligolactic acid **TCA-28** in cone, partial cone and 1,3-alternate conformations has allowed to obtain hybrid nanomaterials, where morphology of silver particles was shown to depend on thiacalixarene’s conformation. Electrochemical reduction of silver ions has allowed to obtain tree-like nanostructured metallic materials. These materials were shown to act as effective electrochemical sensors [99]. The SEM (scanning electron microscopy) micrographs show the formation of spherical particles mostly in contact with each other and amalgamated into submicron-sized structures. Electrostatic assembling of DNA on films allows the detection of specific interactions. Silver dendrites deposited on films offer detection of cholinesterase substrate. These nanomaterials were used in electrochemical sensors, allowing to detect 0.1 to 100 μM of tryptophan with the limit of detection down to 0.03 μM [100]. No interference with oxidation of other amino acids (phenylalanine, histidine, cysteine and tyrosine) was found. The electrochemical sensor developed was validated in the determination of tryptophan sedative medication “Formula of calmness” in the presence of vitamins B5 and B6.

Summing up the discussed material, thiacalixarenes are unique building blocks for supramolecular assembly with silver ions, easily allowing to create three-component assemblies. Thiacalixarenes, due to efficient interaction with silver ions, are very promising stabilizing agents for silver nanomaterials, which would act both as nanostructure-directing agents and stabilizers for prolonged performance of silver nanoparticle (or nanocluster)-based devices.

## 4. Synthesis and Application of Resorcin[4]arene-Based AgNPs

Resorcinarenes are different from other metacyclophanes because of their possibility of functionalizing bridge fragments (forming “lower rim”) by using functional aldehydes at the macrocyclization stage of their synthesis. Presence of eight OH groups forming the upper rim and absence of diverse conformations make their host–guest properties quite specific: they tend to form dimeric capsules enclosing the target substrate due to interactions between eight substituents of each cyclophane molecule [101]. It is especially useful for colloidal plasmonic sensors in which SERS is dependent on nanoparticle aggregation, and in the presence of the target substrate the capsule is formed, which leads to nanoparticle aggregation [102]. (Thia)crown derivatives of resorcinarene are an exception to this rule: such functionalities distort resorcinarene symmetry and present efficient cation binding functional groups. Therefore, such resorcinarene derivatives tend to form crystalline structures, which can differ depending on solvent or thiacrown fragment structure [103], and are out of the scope of our review. To the best of our knowledge, there are no reports on particles obtained by supramolecular self-assembly of resorcinarene derivatives with silver ions.

Similarly to calixarenes and thiacalixarenes, sulfonate derivatives of resorcinarenes can be used for modification of AgNPs (Table 1). Modification of AgNPs (d = 38 ± 5 nm) with *p*-sulfonatoresorcinarene **RA-1** (Figure 7) was reported leading to the formation of larger (d = 45 ± 10 nm) particles that aggregate in the presence of dimethoate (leading to formation of submicron-sized associates, d = 734 nm) [104]. Selective recognition of dimethoate among structurally close dichlorvos, parathion, 2,4-D, dimethoate, hexaconazole, imidacloprid and monocrotophos, was demonstrated.

The presence of eight spatially preorganized functional groups can lead to interesting association behaviors of resorcinarenes. Resorcinarene **RA-2** functionalized with amidoethlyamine fragments has an affinity to the silver nanoparticles surface (which is unusual for amines), which allows to use its self-associates as a supramolecular template for creating hybrid nanomaterial with AgNPs [105]. The authors describe using submicron-sized tubular self-associates of **RA-2** for self-assembly with silver nanoparticles (d = 20.9 ± 16.4 nm). Such assembly is reversible: treatment of resulting hybrid nanomaterials with ultrasound leads to release of AgNPs.

While it is known that substrate encapsulation is often accompanied by formation of dimeric capsules of resorcinarenes, it should be noted that aggregation of resorcinarene molecules on the surface of AgNPs also can assist encapsulation. Such an effect was reported for **RA-3** (derivative of **RA-2** functionalized with 20 carboxylate fragments) [106]: efficiency of doxorubicin binding significantly increased in the case of **RA-3** associated on AgNPs surface, compared to free **RA-3** (95% and 82% of doxorubicin was bound at equimolar **RA-3**:doxorubicin by **RA-3**-AgNPs and **RA-3** solution, respectively). Also, it should be noted that **RA-3** acts as surface-stabilizer of AgNPs synthesized via the sodium borohydride reduction route in the presence of **RA-3,** leading to formation of small-sized nanoparticles (2–3 nm as defined by TEM).

Octacarboxylate derivatives of resorcinarene **RA-4** were also successfully used as a stabilizer in silver ion reduction [107]. **RA-4** increases the affinity of AgNPs formed to the cationic surfactant cetyltrimethylammonium bromide, acting both as host and counter ion. This effect allows to extract nanoparticles into chloroform phase from water.

In a series of reports by Prof. V. K. Jain’s group, hydrazide derivatives of resorcinarene were used for reducing silver salts to AgNPs. An octahydrazide derivative of resorcinarene **RA-5** was successfully used to reduce silver nitrate [108]. Obtained luminescent nanoparticles (d = 7 ± 1 nm) showed pronounced antibacterial activity and were stable in wide range of temperatures (10–50 °C) and pH (4–10). Dispersions were stable at room temperature for 120 days. Also, selectivity towards Fe^3+^ ions was shown (over Zn^2+^, Cd^2+^, Hg^2+^, Co^2+^, Cu^2+^, Pb^2+^, Cr^3+^ and V^3+^). Antibacterial activity was studied on *Escherichia coli*, *Bacillus megaterium, Staphylococcus aureus* and *Bacillus subtilis*. The authors explained the antibacterial activity by a two-step mechanism: adsorption of AgNPs to the cell wall and diffusion of silver ions through cell walls.

Resorcinarene **RA-6** was used for reducing silver ions: four hydrazide moieties in bridge fragments acted as reducer [109]. Resulting AgNPs (d = 5 ± 2 nm) have selectivity towards Cd^2+^. Substitution of eight methoxy groups in **RA-6** with hydroxyl fragments in compound **RA-7** switched AgNPs selectivity [110]. **RA-7**-AgNPs (d = 15 ± 5 nm) selective to Pb^2+^ ions were obtained.

Selectivity towards metal ions was not only demonstrated for AgNPs resulting from reducing Ag^+^ with hydrazide derivatives of resorcinarenes. For example, AgNPs (7 ± 5 nm) were obtained using dodecahydrazide derivatives of resorcinarene **RA-8** as a reducing agent for AgNO_3_. [111]. Obtained **RA-8** AgNPs are sensitive towards histidine in the concentration range of 10 nM–10 μM. **RA-8** AgNPs interact with DNA and show free radical scavenging activity.

Not only hydrazide derivatives of resorcinarene can be used for reducing silver salts. Ferrocene derivatives of resorcinarene **RA-9** form associates which are capable of reducing silver ions into AgNPs [112]. Also, presence of ferrocene fragments provides nanoparticles with catalytic activity in oxidizing *p*-aminophenol to *p*-nitrophenol. Obtained AgNPs have a diameter of 30 nm according to TEM. The thick shell of resorcinarene leads to differences among TEM-defined sizes, and the diameter was determined by AFM and DLS (~60 nm).

In summing up this section, while resorcinarenes have similar cavity shapes, the selectivity of resorcinarenes can be easily adjusted by varying upper rim substituents and bridge fragments. The tendency to form dimeric capsules in the presence of a substrate is extremely favorable for creating AgNPs-based colorimetric sensors. While a significant fraction of resorcinarene derivatives were used for direct synthesis of AgNPs from silver ions, this should not be regarded as a feature specific to resorcinarene derivatives, and most likely there are no obstacles to using similar strategies or similar substituents with other cyclophanes. The use of resorcinarene-based AgNPs mainly consists in the selective recognition of cations and biomolecules in an aqueous solution.

## 5. Supramolecular Self-Assembly of Pillar[n]arenes with Ag(I) Ions and Pillar[n]arene-Based AgNPs

Pillar[n]arenes are a relatively novel class of *para*-cyclophanes, which was first described in 2008 [113]. These macrocycles attract special interest of researchers due to possibility of creating unique supramolecular systems based on them [114]. Pillararenes have a number of attractive features, such as synthetic accessibility, planar chirality and a tube-shaped three-dimensional structure, which provides an electron-donor cavity and the possibility of stringing elongated acyclic and planar cyclic guest fragments. Due to these unique properties, developing novel functional organo-inorganic nanomaterials based on pillararenes is especially attractive [115]. The key feature of pillar[5]arenes compared to other cyclophanes is the combination of a wide range of synthetically accessible functional fragments inherent for calixarenes and resorcinarenes with the possibility of forming pseudo-rotaxanes, rotaxanes and poly-pseudo-rotaxanes [116,117]. The presence of metal ions in supramolecular architectures allows to adjust their properties and provides them with unique optical, magnetic and electric properties [118].

Therefore, substrate recognition in the case of pillararenes strongly depends on substrate geometry: a higher affinity is observed if the substrate fits inside the cavity of pillararene. While using aggregation-induced shift of the absorption band in substrate recognition is a common practice for creating selective AgNP-based sensors, in the case of pillararenes the pattern of interaction with substrate can be in sharp contrast to other cyclophanes. For the other abovementioned cyclophanes ((thia)calixarenes, resorcinarenes), the substrate fits inside the cyclophane binding site or molecular cage formed by several cyclophane molecules (e.g., dimeric capsules of resorcinarenes). On the contrary, in the case of pillararene–AgNPs, interaction with substrate may effectively induce AgNPs aggregation even if only some of its fragments are “inserted” into pillar-shaped cavities of pillararenes (Table 1).

### 5.1. Self-Assembly of Pillar[n]arenes with Ag(I) Ions

There are some publications that report formation of pillararene-Ag^+^ host–guest complexes [119]. Monosubstituted pillar[5]arene with a pyridine group **PA-1** (Figure 8) is capable of forming dimers in the presence of silver ions, forming supramolecular polymer in the presence of a homoditopic guest [120]. The polymer obtained can reversibly bind H_2_S or I^−^ in chloroform. The authors suppose that the proposed material can be used for creating advanced sensor materials.

1D-coordinated polymer based on disubstituted pillar[5]arene **PA-2** containing thiopyridyl moieties and silver cations was formed [121]. Upon addition of α,ω-dicyanoalkanes [CN(CH_2_)_n_CN, n = 2–8] (Figure 9) to that supramolecular system, 1D and 2D poly-pseudo-rotaxanes are formed (Figure 10). The length of α,ω-dicyanoalkane significantly affects the structures of corresponding polypseudorotoxanes: 1D poly-pseudo-rotaxane is formed with CN(CH_2_)_2_CN (**G1**). In the case of CN(CH_2_)_8_CN (**G2**), 2D poly-pseudo-rotaxane is formed, in which the same guest is forming both threads and crosslinks. The authors report that results obtained allow to create poly-pseudo-rotaxane based inorganic/organic hybrid materials, incorporating macrocyclic components.

Amphililic pillar[5]arene containing carboxylate charged fragments on one site of the macrocyclic rim and lipophilic pentyl groups **PA-3** does not form associates in water, but adding silver ions leads to dendrite structures [122]. Stoichiometry, structure, sizes and morphology of associates formed are studied by a complex of methods (DLS, TEM, SEM).

Unfortunately, there are few publications on the self-assembly of pillar[n]arenes and silver cations, perhaps due to the fact that pillararenes are a new class of *para*-cyclophanes. However, the possibility of pillararenes to form poly-pseudo-rotaxane structures with silver cations opens up prospects for the creation of new materials with unique physical properties.

### 5.2. Synthesis and Application of Pillar[n]arene-Based AgNPs

Pillar[n]arene-modified AgNPs combine electronic, thermal and catalytic properties of metal nanoparticles with the ability of molecular recognition of various guests fitting into the pillararene cavity (Figure 9). The tendency to form complexes with substrates having long fragments fitting into the pillararene cavity was described above. In a fashion similar to individual pillararene molecules, if the substrate has several such fragments, formation of the complex results in aggregation of AgNPs.

In Prof. M. Xue’s group, AgNPs modified by decacarboxylate pillararene **PA-4** and dodecacarboxylate pillar[6]arene **PA-5** were synthesized. Synthesis of AgNPs stabilized with a water-soluble derivative of pillar[5]arene **PA-4** with narrow size distribution (18.7 ± 2.18 nm) was reported [123]. Nanoparticles were obtained by reducing silver nitrate (10 mM) with NaBH_4_ (50 mM) in the presence of varying concentrations of **PA-4**. Optimal concentration of **PA-4** (0.4 mM) was defined. Obtained nanoparticles were stable at high temperatures over long time periods. AgNPs interact with spermine **G3** and its structural analogues leading to aggregation-induced sedimentation. Pillararene **PA-4**-based AgNPs are capable of interacting with water-soluble guest molecules containing two paraquate fragments **G4** [124]. Adding of dodecacarboxylate **PA-5** leads to the concurrent process of guest binding with **PA-5,** which leads to disaggregation of **PA-4**-stabilized AgNPs. The authors show that disaggregated nanoparticles can be used repeatedly: centrifugation allows to separate **PA-4**-stabilized AgNPs and to redisperse them in water.

Further studies with decacarboxylate derivatives of **PA-4-**stabilized **AgNPs** are reported [125], and an electrochemical sensor has been developed, allowing detection of paraquate **G5** (PQ) herbicide (highly toxic for humans). In this work, **PA-4**-stabilized AgNPs were immobilized on graphene surface. Glassy carbon electrodes modified with hybrid material obtained demonstrate pronounced sensitivity towards PQ. They show good current response toward PQ, with limit of detection for PQ of 1.0 × 10^−8^ M. Such synergism is explained by complementarity of the pillar[5]arene cavity towards PQ.

A green and facile approach towards synthesis of small AgNPs (3–4 nm) on a single-walled carbon nanotube surface (SWCNT) modified with pillar[6]arene **PA-6** containing 12 phosphate groups was described [126]. Nanoparticles were obtained by adding aqueous solution of AgNO_3_ (10.0 mM) to a dispersion of **PA-6**-SWCNT, following with slow addition of NaBH_4_ (0.1 M). Obtained AgNPs show higher activity towards reduction of 4-nitrophenol and methylene blue degradation compared to analogous catalysts. Highly sensitive electrochemical sensors based on obtained nanoparticles for PQ detection has been developed. The authors report these results as a promising approach to create highly effective catalytic nanomaterials for organic dye degradation and detection of highly toxic herbicides.

The same research group obtained AgNPs functionalized with pillar[6]arene **PA-5** containing 12 carboxylate groups [127], which were cast on 2D covalent organic framework (COF) composite. The obtained hybrid material has high sensitivity towards PQ. Electrodes based on this material can detect 0.01–50 μM PQ with a detection limit of 0.014 μM.

AgNPs stabilized with water-soluble pillar[5]arene **PA-7** and pillar[6]arene **PA-8** containing 10 and 12 imidazolium fragments were obtained [128]. The concentration ratio of **PA-8** to AgNO_3_ varied, the optimal ratio was [**PA-8**]/[AgNO_3_] = 0.15, at which AgNPs size was 13.57 ± 2.18 nm. It was shown that **PA-8**-stabilized AgNPs can be used as colorimetric sensor for selective detection of glutamic acid **G6** in water (in a row of lysine, arginine, histidine, glycine, glutamic acid, tyrosine, aspartic acid and threonine). In the presence of glutamic acid, the colloid system color changes from yellow to red (detection limit = 2.8 × 10^−6^ M). It should be noted that the pillar[5]arene derivative **PA-7** does not show such selectivity. This work is a good illustration of the pillararene-specific pattern of substrate recognition: substrate (or its fragment) should “fit” in the pillararene cavity for selective and efficient recognition.

Silver nanoclusters were modified by monosubstituted pillar[5]arene **PA-9** with fragments of lipolic acid obtained both by straight synthesis in the presence of **PA-9** and by ligand exchange [129]. It was shown that they were stable at room temperature over a four-month period. Obtained nanoclusters interact with neutral (alkylamine) and charged quaternary ammonium (**G7)** guest molecules, which lead to the formation of spherical aggregates with unique optical properties. The authors report that while alkylamines produced a 30-fold photoluminescence enhancement, positively charged quaternary ammonium molecules induced an approximately 2000-fold photoluminescence enhancement that can be perceived by the naked eye.

For summing up the publications mentioned, it is interesting to compare pillararene-functionalized AgNPs to AgNPs modified with other cyclophanes, in contrast to (thia)calixarenes: complexation with guest molecule, to a larger extent, is guided by “fitting” guest molecule fragments into the macrocyclic cavity rather than by the nature and number of substituents. In contrast to resorcinarenes where substrates should usually fit into the cavity of dimeric capsules to induce AgNPs aggregation, for pillararenes it is enough to form inclusion complexes incorporating fragments of the substrate inside pillararene cavities. It would be intriguing to see multisensor systems based on these different patterns which would allow to effectively discern over a wide range of structurally related substrates. While AgNPs functionalized with resorcinarene derivatives would most likely bind those guest molecules which are best fit for dimeric capsules, pillararene-functionalized AgNPs would more effectively aggregate in the presence of guest molecules containing alkyl and aryl groups, fitting inside cavities of single pillararene molecules.

## 6. Conclusions

This review summarizes recent progress in self-assembly of cyclophanes with silver cations and in the design, synthesis and application of cyclophane-based silver nanoparticles until 2020. Various strategies are used to create silver nanoparticles such as modifying, functionalizing, templating, reducing and stabilizing. As mentioned in the above publications, the choice of functional groups of cyclophanes for modification of AgNPs is quite similar. On the other hand, each class of macrocycles (calix[n]arene, thiacalix[4]arene, resorcin[4]arene, pillar[n]arene) offers unique patterns of interacting with the substrate. For example, significant differences in organic substrates that are recognized by resorcinarene-modified AgNPs and pillararene-modified AgNPs are observed. Thiacalixarenes prove promising for Ag nanocluster stabilization and for creating supramolecular ensembles with silver ions, which encourages their use in supramolecular templating of AgNPs formed by reduction.

The obtained nanoparticles are used for recognition of various biomolecules in aqueous media, namely amino acids, hydroxyl acids, proteins and nucleic acids. Based on cyclophane-capped silver nanoparticles, electrochemical and colorimetric sensors were created for the selective determination of heavy metal cations (Cu^2+^, Fe^3+^, Hg^2+^, Cd^2+^, Pb^2+^), anions (H_2_PO_4_^−^, I^−^), amino acids (histidine, aspartic acid, tryptophan, glutamic acid), polycyclic aromatic hydrocarbons and pesticides. Therefore, we can conclude that over the last years cyclophanes have opened new and interesting opportunities for selective substrate recognition (based on unique patterns of substrate-induced aggregation). Depending on substrate structure, cyclophane type and functionality can be successfully devised. While it was shown that silver nanoparticles can be successfully prepared in the presence of any cyclophane mentioned above, it is still intriguing to study the relation between silver ion aggregation with cyclophane and the sizes of AgNPs formed. Unfortunately, this field is still waiting to bloom.

It is certain that cyclophane-based silver nanoparticles will find application in many areas of life. Cyclophane-functionalized AgNPs have proven promising for medicine and diagnostic fields. The tendency of cyclophanes to form supramolecular assemblies with various substrates allows using them in constructing AgNP-based colorimetric sensors due to substrate-induced nanoparticle aggregation. Depending on the macrocyclic platform and functionalization, cyclophanes are selective to a wide range of substrates. This possibility of adjusting association behavior ranges from small inorganic and organic substrates, and allows creating AgNPs with adjustable cytotoxicity towards different bacteria. A different pattern is implemented in electrochemical sensors: differences in affinity towards substrates allow achieving a selective response towards small-sized substrates. The use of cyclophanes for silver nanocluster stabilization is emerging. Also, there are future prospects for use of cyclophane–silver ion assemblies as templates for Ag(0) nanomaterial designs due to manifold morphologies of associates formed with silver ions. It can be concluded that, in the nearest future, cyclophane-modified AgNPs and silver nanomaterials will find applications in nanoelectronics.

## Figures and Tables

**Figure 1 ijms-21-01425-f001:**
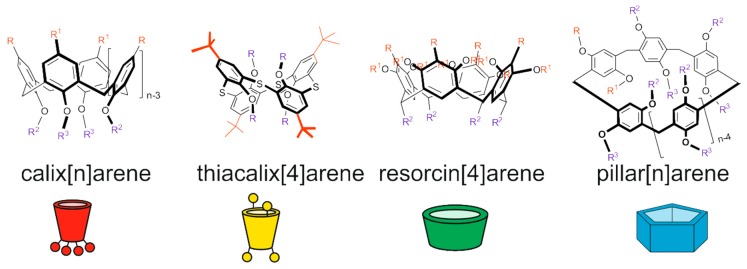
Structures and cartoon images of cyclophanes (calix[n]arenes, thiacalix[4]arenes, resorcin[4]arenes and pillar[n]arenes).

**Figure 2 ijms-21-01425-f002:**
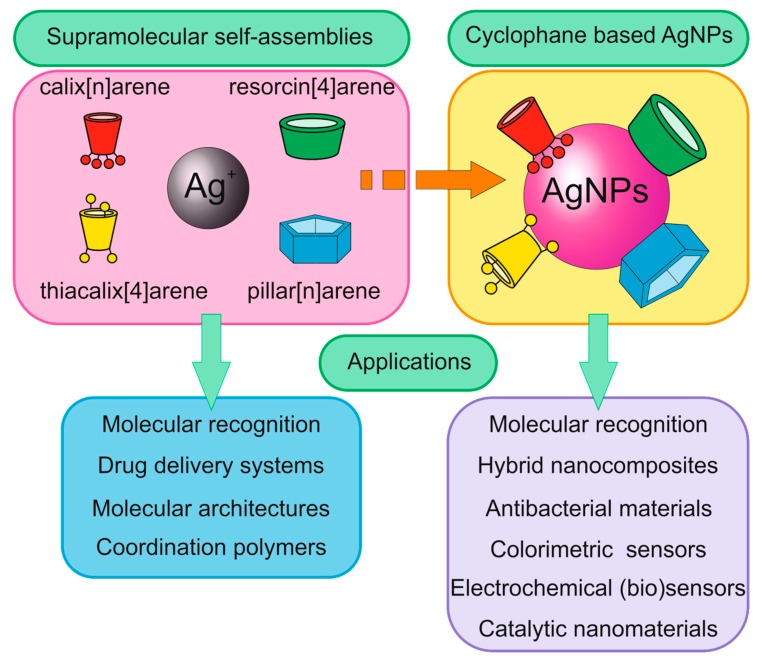
Application of supramolecular self-assemblies “cyclophane-Ag^+^” and cyclophane based silver nanoparticles (AgNPs).

**Figure 3 ijms-21-01425-f003:**
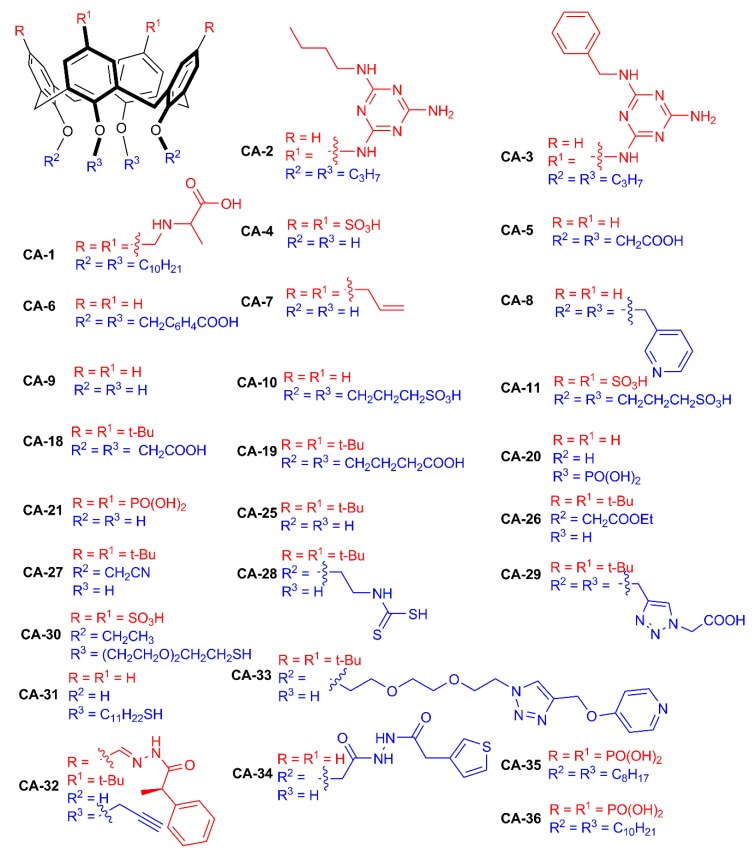
Structures of calix[4]arenes.

**Figure 4 ijms-21-01425-f004:**
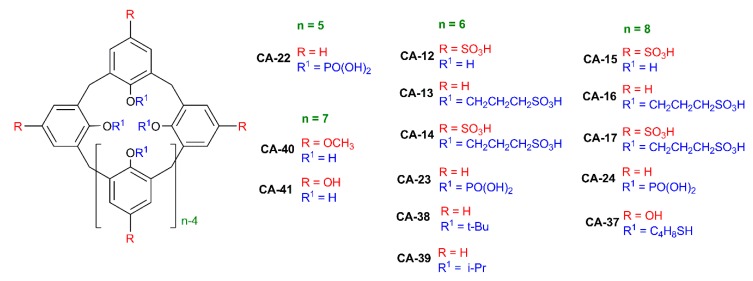
Structures of calix[n]arenes (n = 5–8).

**Figure 5 ijms-21-01425-f005:**
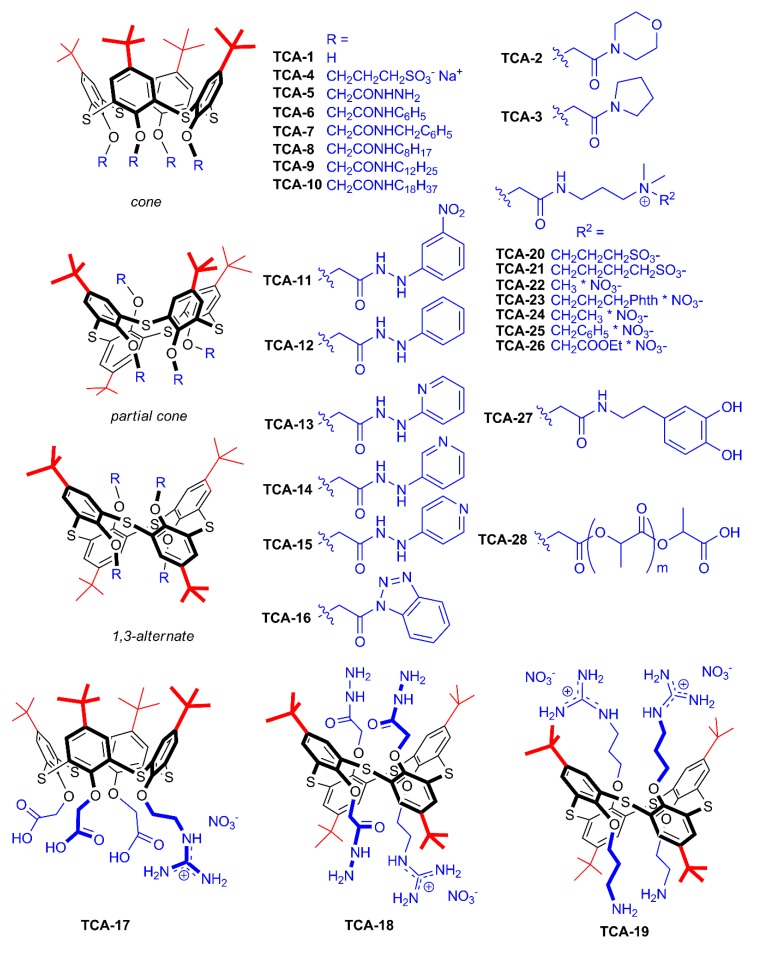
Structures of thiacalix[4]arenes.

**Figure 6 ijms-21-01425-f006:**
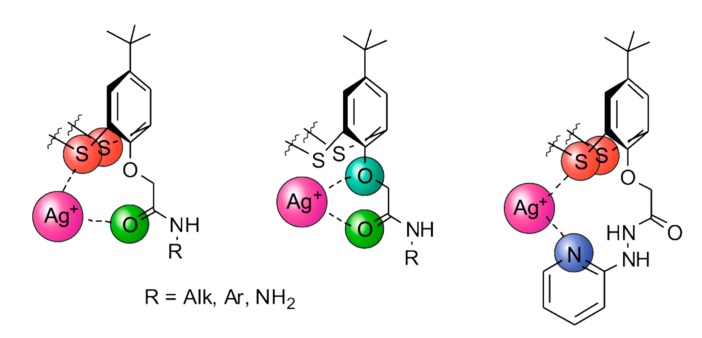
Possible coordination of silver ions with amide derivatives of thiacalix[4]arene.

**Figure 7 ijms-21-01425-f007:**
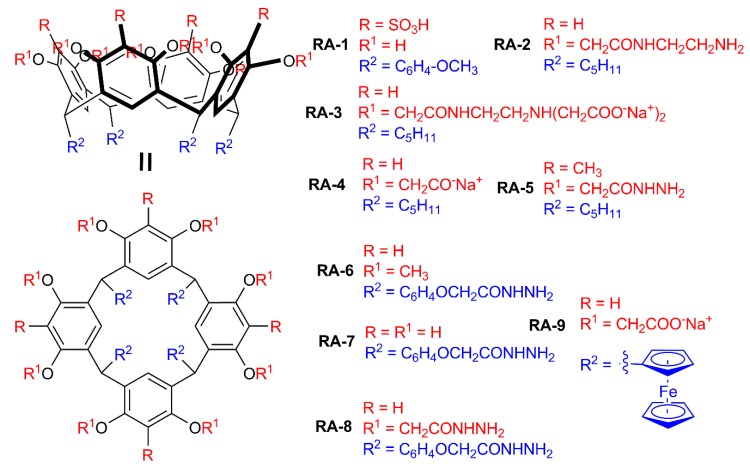
Structures of resorcin[4]arenes.

**Figure 8 ijms-21-01425-f008:**
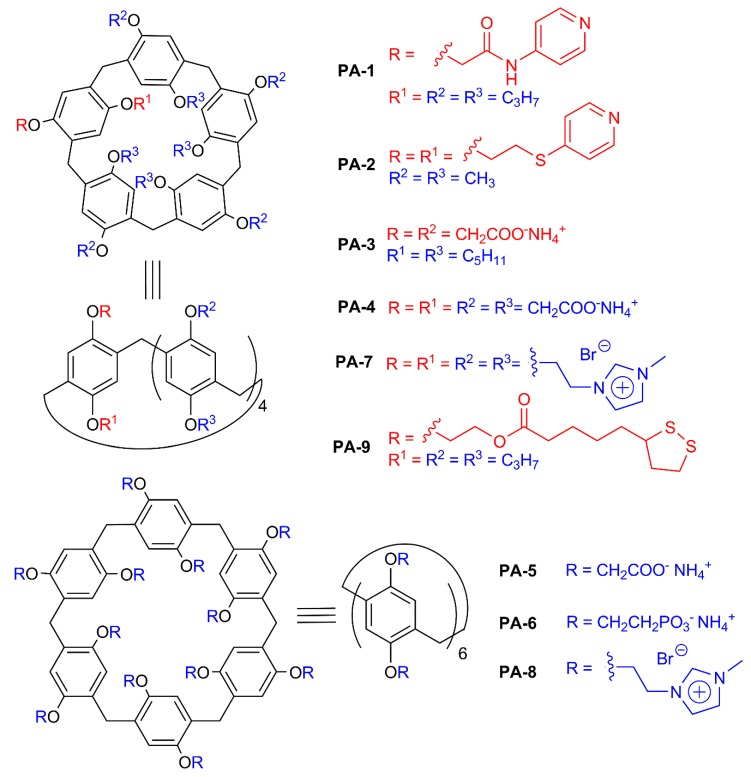
Structures of pillar[n]arenes.

**Figure 9 ijms-21-01425-f009:**
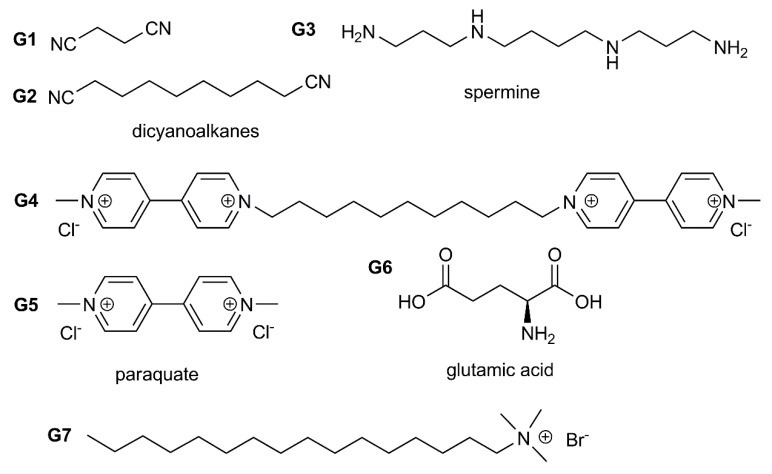
Structures of guest molecules for recognition using pillar[n]arene-based AgNPs.

**Figure 10 ijms-21-01425-f010:**
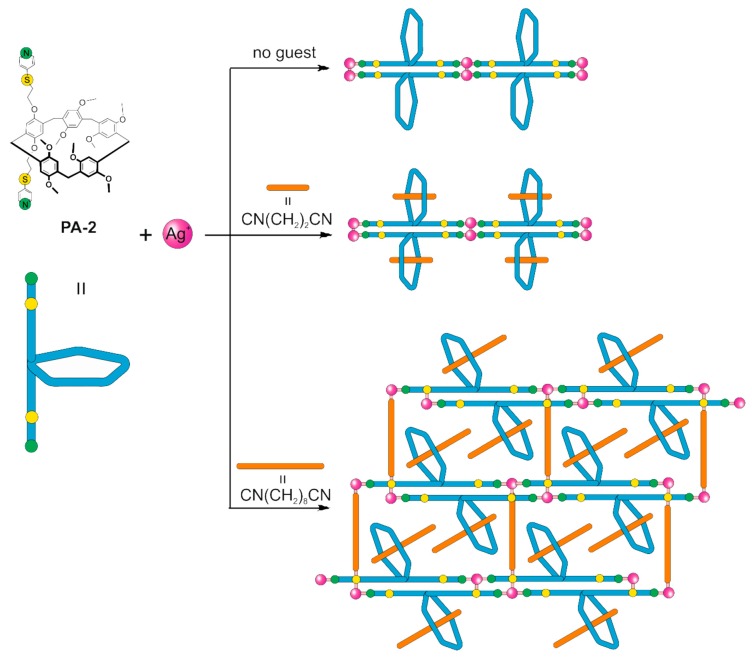
Cartoon representation of molecular architectures obtained by self-assembly of pillar[5]arene **PA-2**, silver ions and **G1** and **G2** guest molecules.

**Table 1 ijms-21-01425-t001:** Data on cyclophane-based silver nanoparticles (AgNPs) sizes, applications, target and comparison substrates.

Cyclophanes Studied	AgNPs Synthesis Method ^1^	AgNPs Sizes (Method)	Surface Plasmon Resonance λ	Application, Target Substrate, SPR λ, nm in the Presence of Substrate ^2^	Comparison Substrates	Lit.
**CA-4, CA-10–CA-17**	B (NaBH_4_)	-	390 nm	CMC determination of cationic surfactants (CPB, CTAB) intensity of 390 nm band **CA-10**—decrease **CA-4**, **CA-11**—increase **CA-12–CA-17**—no changes	cetyl pyridium bromide, cetyl trimethyl ammonium bromide, *N*-octyl glucopyranoside.	[34]
**CA-4, CA-10–CA-17**	B (NaBH_4_)	-	-	Structure-dependent inhibition of Gram+ and Gram− bacteria growth, anti-oxidant capacity, antibacterial effect against *E. coli, B. subtilis*	-	[35,36]
**CA-4**	B (NaBH_4_)	~20 nm (TEM)	~380 nm	Multicolor response to nucleotides and desoxynucleotides	cytosine, guanine—540 nm, uracil—580 nm, thymine—590 nm, adenine—no changes, deoxy-adenosine, deoxy-guanosine—460 nm, deoxy-thymidine—520 nm, deoxy-cytidine—decoloration (grey colored solution)	[37]
**CA-2, CA-3**	B (electron beam)	**CA-2**: 2.3 ± 0.3 nm (TEM) **CA-3**: 2.1 ± 0.1 nm (TEM)	-	-	-	[26]
**CA-4**	B (NaBH_4_)	~30 nm (TEM)	390 nm	540 nm (cytosine)	-	[38]
**CA-4**	A (NaBH_4_)	11.6 ± 3 nm (TEM)	407 nm (hexane) oleic acid stabilized	413 nm (water) oleic acid stabilized+calixarene; Oleic-stabilized particles become hydrophilic	-	[39]
**CA-4**	B (NaBH_4_)	8.0 ± 1.0 nm (TEM)	394 nm	His ~ 500 nm	Alanine, valine, leucine, methionine, phenylalanine, histidine tyrosine, threonine, serine, proline, glutamic acid, aspartic acid	[40]
**CA-4**	B (NaBH_4_)	~6–12 nm (TEM)	391 nm	450–600 nm (transition metal hydroxide particles at pH = 10)	Effectivity is: Ni^2 +^ <Tb^3 +^ ~Zn^2 +^ <Cu^2 +^ <Co^2+^~Cd^2+^ << Pb^2+^	[41]
**CA-4, CA-15**	A (NaBH_4_)	**CA-4**: 8 nm (TEM) **CA-15**: 4 nm (TEM)	**CA-4**: 393 nm **CA-15**: 391 nm	**CA-4** Optunal ~ 500 nm **CA-15 -**	iprodione, pyrimethanil, thiabendazole, optunal, parathion-methyl, methomyl and acetamiprid	[42]
**CA-12**	A (NaBH_4_)	-	-	Three-component supramolecular system with dipyrene, discriminates; H_2_PO_4_^-^	NaF, NaCl, NaBr, NaI, NaH_2_PO_4_, NaHSO_3_, Na_2_SO_4_, NaNO_3_, NaNO_2_ and NaHCO_3_	[43]
**CA-12**	B (KOH,400º C, H_2_)	14.9 ±6.7 nm – AgNPs (TEM) 15.6 ± 9.1 nm – graphene nanocomposites (TEM)	-	Antibacterial activity against *S. aureus, E. Coli*	-	[44]
**CA-4, CA-12, CA-15**	B (NaBH_4_)	**CA-4**: 4.5 nm (TEM) **CA-12**: 5 nm (TEM) **CA-15**: 8 nm (TEM)	400 nm 400 nm 400 nm	415 nm 410, 465 nm 410, 450, 490 nm (Saburopin)	increased Saburopin efficiency and stability.	[45]
**CA-4, CA-10, CA-11, CA-18, CA-19, CA-20**	B(NaBH_4_)	-	**CA-4**: 390 nm **CA-10**: 390 nm **CA-11**: 380 nm **CA-18**: 420 nm **CA-19**: 400 nm **CA-20**: 400 nm	Chlorohexidine, gentamycine	chlorohexidine, chloramphenicol, gentamycine sulfate,	[46]
**CA-4, CA-20**	B(NaBH_4_)	-	**CA-4**: 390 nm **CA-20**: 398 nm	**CA-4**: 398 nm (BSA), **CA-20**: 404 nm (BSA)	-	[47]
**CA-21 CA-22 CA-23 CA-24**	B (H_2_) (pH = 9) 70 °C	**CA-21**: 2.1± 0.8 nm (TEM) **CA-22**: 2.9 ±1.3 nm (TEM) **CA-23**: 5 ± 3.6 nm (TEM) **CA-24**: 5.3± 2.4 nm (TEM)	~ 350, 390 nm	-	-	[48]
**CA-25**, **CA-26**, **CA-27**	A (NH_2_OH·HCl)	~23 nm (TEM)	420 nm	436 nm (pyrene)	pyrene, triphenylene, benzo[c]phenantrene, anthracene, coronene, chrysene, dibenzoanthracene, rubicene	[50]
**CA-28**	A (sodium citrate or NH_2_OH)	-	-	SERS (coronene)	pyrene, triphenylene, benzo[c]phenantrene, coronene	[51]
**CA-25**	B (ethylene-diamine)	9.4 nm (nanocrystalline) (TEM)	-	-	-	[54]
**CA-12**	B (NaBH_4_)	7 nm (TEM)	393 nm	406 nm (Sanguinarine)	Enhancement of antibacterial activity	[55]
**CA-29**	A (NaBH_4_, C_18_H_37_NH_2_)	46.7 nm (DLS)	430 nm	430 nm (intensity increase)	CoCl_2_, NiCl_2_, MnCl_2_, Cd(NO_3_)_2_, AgNO_3_, Cu(NO_3_)_2_), Pb(NO_3_)_2_, Hg(NO_3_)_2_, BaCl_2_	[56]
**CA-30**	A (NaBH_4_)	52 nm (TEM)	422 nm	554 nm (Fe^3+^)	Zn Fe^3+^, Cu^2+^, Ca^2+^, Co^2+^, Mg^2+^, Cd^2+^, Ba^2+^, Na^+^, K^+^, Mn^2+^, Fe^2+^, Pb^2+^, Ni^2+^, Pd^2+^, Hg^2+^, pepsin, cytochrome c, BSA, myoglobin	[57]
**CA-31**	A (interphase NaBH_4_, C_12_H_25_SH)	3 ± 1 nm (TEM)	450 nm	580 nm (association with AuNPs)	AuNPs functionalized with pyridinium fragments	[58]
**CA-33**	B (NaBH_4_)	9 nm (TEM)	~ 415 nm	~ 525 nm (N-Fmoc-L-aspartic acid)	N-Fmoc-L-aspartic acid, N-Fmoc-D-aspartic acid	[59]
**CA-32**	B (photo-reduction, 365 nm)	10 ± 1.0 nm (TEM)	364 nm	Fe^3+^ (414 nm)	Li^+^, Na^+^, K^+^, Cs^+^, Mg^2+^, Ca^2+^, Sr^2+^, Ba^2+^, Cr^3+^, Fe^3+^, Fe^2+^, Cu^2+^, Pb^2+^, Ag+, Ni^2+^, Mn^2+^, Co^2+^, Cd^2+^, Zn^2+^.	[60]
**CA-34**	B (photo-reduction with sunlight)	3-5 nm (TEM)	432 nm	432 nm, lower intensity (Hg^2+^)	Li^+^, Na^+^, K^+^, Cs^+^, Ca^2+^, Mg^2+^, Ba^2+^, Cr^3+^, Sr^2+^, Co^2+^, Ni^2+^, Cd^2+^, Zn^2+^, Rb^+^, Hg^2+^, Pb^2+^, Cu^2+^	[61]
**CA-21, CA-23, CA-24, CA-35, CA-36**	B (photo-reduction, pH = 9, 365 nm)	**CA-21**: 3.6 ± 1.2 nm **CA-23**: 2.6 ± 0.8 nm **CA-24**: 2.6 ± 0.8 nm **CA-35**: 3.6 ± 1.2 nm **CA-36**: 3.6 ± 1.2 nm	**CA-21**: 400 nm **CA-23**: 400,485 nm **CA-24**: 400,485 nm **CA-35**: 400 nm **CA-36**: 400 nm	-	-	[62]
**CA-37**	B (gamma-irradiation)	1.55–0.5 nm in order AgPF_6_ < AgClO_4_ < AgOTf < AgBF_4_ Bimetallic: 1 nm and 3 nm (HAADF-EDX-STEM)	410 nm	nitrothiophenol reduction	nitrothiophenol, nitrophenol	[63]
**CA-12**	B (electrochemical reduction)	~ 100 nm (TEM)	-	methyl parathion	PO_4_^3−^, SO_4_^2−^, CO_3_^2−^, NO^3−^, *p*-nitrophenol, nitrobenzene	[64]
**CA-25**	B (electrochemical reduction)	40–70 nm (TEM)	-	nifedipine	dopamine, ascorbic acid, L-Dopa, epinephrine, tryptophan, L-cysteine, uric acid	[65]
**CA-25, CA-38**	B (electrochemical reduction)	~ 100–2000 nm (TEM)	-	flutamide	dopamine, ascorbic acid, L-Dopa, epinephrine, tryptophan, L-cysteine, uric acid	[66]
**CA-39**	B (electrochemical reduction)	70 nm (TEM)	-	electrochemical reduction of H_2_O_2_	-	[67]
**CA-40, CA-41**	Combination of C and B (electrochemical reduction with participation of cyclophane as reducing agent)	10 nm, height = 2 nm (AFM)	-	Photocatalytic degradation of methyl orange, methylene blue, rhodamine 6G chloride	-	[68,69]
**TCA-3, TCA-8**	B (reduction with DMF)	three-dimensional self-assembled monolayer (DMF) (1–13 nm; 46–622 nm	-	-	-	[92]
**TCA-5**	C	20 nm	~425 nm	~ 452 nm (histidine) ~414 nm (tryptophan)	valine, proline, arginine, cysteine, aspartic acid, glutamic acid, glutamine, leucine, methionine, phenylalanine, tryptophan, isoleucine, histidine	[93]
**TCA-1**	A	Two nanoclusters with 35 and 34 atoms	Ag36: 501, 336, 300 (shoulder) nm Ag35: 495, 336, 300 (shoulder) nm Ag34: 482, 336, 300 (shoulder) nm	-	-	[94]
**TCA-27**	C	4–6 nm (TEM)	-	Substrate depending on electrode composition: A: dopamine B: ochratoxin C: cholinesterase D: DNA damage	-	[95,96,97,98]
**TCA-28**	B (electrochemical reduction)	Micro-sized, contacted granules (TEM)	-	A: factors affecting DNA charge and structure (thermal denaturing, methylene blue intercalation, oxidative damage), thiocholline B: tryptophan	B: phenylalanine, histidine, cysteine, tyrosine	[99,100]
**RA-1**	A	45 ± 10 nm (DLS)	420 nm	519 nm (dimethoate)	dichlorvos, parathion, 2,4-D, dimethoate, hexaconazole, imidacloprid and monocrotophos	[104]
**RA-2**	A	20.9 ± 16.4 nm (TEM)	416 nm	reversible association with resorcinarene self-assembled microtubes	-	[105]
**RA-3**	B (NaBH_4_)	2–3 nm (TEM)	-	doxorubicin	-	[106]
**RA-4**	B (NaBH_4_)	4–6 nm (TEM)	429 nm	Cetyltrimethylammonium (phase transfer of AgNPs from water to chloroform upon interaction with surfactant)	-	[107]
**RA-5**	C	7 ± 1 nm (TEM)	406 nm	Fe^3+^ (fluorescence quenching at 560 nm), antibacterial activity against *E. coli*, *B. subtilis*, *S. aureus*, *B. megaterium*	Zn^2+^, Cd^2+^, Hg^2+^, Co^2+^, Cu^2+^, Pb^2+^, Cr^3+^, V^3+^	[108]
**RA-6**	C	5 ± 2 nm (TEM)	415 nm	Cd^2+^ (fluorescence quenching at 458 nm)	Zn^2+^, Pb^2+^, Co^2+^, Cu^2+^, Ba^2+^, Cd^2+^, Mn^2+^, Hg^2+^, Ca^2+^, Mg^2+^, Sr^2+^, Ni^2+^	[109]
**RA7**	C	15 ± 5 nm (TEM)	426 nm	Pb^2+^ (fluorescence quenching at 580 nm)	Cr^3+^, Mn^2+^, Fe^3+^, Co^2+^, Ni^2+^, Cu^2+^, Zn^2+^, Cd^2+^, Hg^2+^	[110]
**RA-8**	C	7 ± 5 nm (TEM)	408 nm	420 nm (CT-DNA, S-DNA) Histidine (fluorescence quenching at 540 nm)	arginine, cysteine, aspartic acid, glutamic acid, glutamine, leucine, methionine, threonine, histidine, L-Dopa, tryptophan	[111]
**RA-9**	C	30 nm (TEM) 60 nm (AFM, DLS)	440 nm	Catalysis of nitrophenol reduction	-	[112]
**PA-5**	B (NaBH_4_)	18.7 ± 2.18 nm (TEM)	400 nm	400 nm intensity decrease (Structural analogues of spermine **G3**)	spermine, ursol, tetraethylenepentamine, triethylenetetramine, ethanediamine, 1,12-dodecylamine and 1,6-hexamethylenediamine	[123]
**PA-4**	B (NaBH_4_)	10 nm (TEM)	404 nm	456 nm (series of alkylidene- linked two paraquat units) binding is reversible upon addition of **PA-5**	-	[124]
**PA-4**	B (NaBH_4_)	~10 nm (TEM)	394 nm	Electrochemical detection of paraquate	-	[125]
**PA-6**	B (NaBH_4_ in the presence of SWCNT)	3–4 nm (TEM)	410 nm	Catalytic reduction of nitrophenol, catalytic degradation of methylene blue	-	[126]
**PA-5**	B (NaBH_4_)	6.01 ± 0.94 nm (TEM)	400 nm	Electrochemical detection of paraquat	-	[127]
**PA-7**, **PA-8**	B (NaBH_4_)	13.57 ± 2.18 nm (TEM)	400 nm	500 nm glutamic acid	lysine, arginine, histidine, glycine, glutamic acid, tyrosine, aspartic acid, threonine	[128]
**PA-9**	A (ligand-exchange), B (synthesis in the presence of cyclophane)	Ag_29_ nanoclusters For case B, formation of small nanoparticles is possible	330, 455, 513, 623, 700 nm	Photoluminiscence enhancement (810 nm, Neutral alkylamines; 650 nm, quaternary alkylammonium salts)	hexylamine, dodecaneamine, oleylamine, 1,8-diaminooctane trimethyloctadecylammonium bromide,	[129]

^1^ A, pre-synthesized; B, reduced in the presence of cyclophane; C, cyclophane is used as reducing agent; ^2^ Application is not specified if AgNPs are used as the colorimetric sensor.

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
