# Peer review of "The Role of Calix[n]arenes and Pillar[n]arenes in the Design of Silver Nanoparticles: Self-Assembly and Application"

_ijms, 2020, doi:10.3390/ijms21041425_

Round 1

Reviewer 1 Report

The authors have done a good jobs in describing the current methods to synthesize and stabilize Ag (0) nanoparticles by self-assembly of associates with Ag (I) ions and the participation of cyclophanes (resorcin[4]arenes, (thia)calix[n]arenes, pillar[n]arenes). The authors not only compared the approaches of syntheses, but also the applications of each groups of AgNPs with cyclophanes. The figures and tables did help readers to understand the description that authors provided.

There are some corrections or suggestions below with the formation as:

Line #,  original sentence  (corrections or suggestions)

  1. Line 39, Depending on the application AgNPs must be  (Depending on the application, AgNPs must be)
  2. Line 42,120, 422, 699, 722 in presence  (in the presence)
  3. Line 43, substrate it is necessary  (substrate, it is necessary)
  4. Line 65, Structure and cartoon image of  (Structures and cartoon images of)
  5. Line 79, practices to obtaining  (practices to obtain)
  6. Line 93,208, 219, 306, 348, 351, 552, 574, 688, 708   Authors  (The authors)
  7. Line 98, 323, 447,537,546, 632, 643 Structure of  (Structures of)
  8. Line 113, have  (has)
  9. Line 119, There is example  (There is an example)
  10. Line 136, in the case of calixarenes the nature and  (in the case of calixarenes, the nature and)
  11. Line 182, with oleic acid stable in  (with oleic acid in)
  12. Line 231, Between  (among)
  13. Line 236, cavity one of  (cavity, one of)
  14. Line 249, coronene the  (coronene, the)
  15. Line 261, of Prof. D. Tian [56] AgNPs  (of Prof. D. Tian [56], AgNPs)
  16. Line 309, in ratio of AgNO3/CA-34 1:15  (in the ratio of AgNO3/CA-34 1:15)
  17. Line 310, studied (20, 50 and 80° C).  (studied at 20, 50 and 80° C.)
  18. Line 319, At pH=9 size of obtained nano  (At pH=9, size of obtained nano)
  19. Line 324, surface let some accessibility  (surface cause (?) some accessibility)
  20. Line328,329, It is interesting, when compared to AgNPs obtained by common chemical reduction route   (what is the “it”? what is compared to AgNPs)
  21. Line 329, shape and size of associates of calixarene  (shapes and sizes of associates of calixarene)
  22. Line 340, On the other hand the   (On the other hand, the)
  23. Line 349, than for   (than those for)
  24. Line 350, in range of 40-70  (in a range of 40-70)
  25. Line 360, show linear signal  (shows linear signal)
  26. Line 398, High affinity  ((Tabbing a space )High affinity)
  27. Line 403, The authors suggest  (The authors suggested)
  28. Line 411, related to forming  (related to form)
  29. Line 423, On the contrast for  (In contrast to )
  30. Line 429, cases conformation  (cases, conformation)
  31. Line 537, our knowledge there are  (our knowledge, there are)
  32. Line 539, and thiacalixarenes sulfonate  (and thiacalixarenes, sulfonate)
  33. Line 568, group hydrazide  (group, hydrazide)
  34. Line 574, и  (and)
  35. Line 583,584, Using dodecahydrazide derivative of resorcinarene RA-8 were obtained AgNPs (7±5 nm) selective towards neutral substrates. (? Not clear)
  36. Line 586, as was shown  (? as shown)
  37. Line 592, difference between TEM defined size  (differences among TEM defined sizes)
  38. Line 609, properties developing  (properties, developing)
  39. Line 664 , group AgNPs modified  (group, AgNPs modified)
  40. Line 670, In same time  (?)
  41. Line 688, to creating  (to create)
  42. Line 690, In the same research group were obtained AgNPs functionalized  (? The same research group obtained AgNPs functionalized)
  43. Line 695, 696, Ratio of PA-8 concentration to concentration of AgNO3 was varied, optimal ratio found was [PA-8]/[AgNO3] = 0.15,  (The concentration ratio of PA-8 to AgNO3 varied, the optimal ratio was [PA-8]/[AgNO3] = 0.15,)
  44. Line 726, 2020 year.  (2020.)
  45. Line 728, As we can see from abovementioned publications  (As mentioned in the above publications)
  46. Line 743, structure cyclophane  (structure, cyclophane)

Author Response

The authors would like to thank the reviewer for the time, positive feedback and valuable recommendations. All corrections suggested by the reviewer have been added.

Reviewer 2 Report

This review presents progress in self-assembly of cyclophanes with silver cations and their application of cyclophane based silver nanoparticles until the 2020 year. The review is comprehensive and acceptable in the journal after few minor comments such as;

The title should be very short and easy to read. Add the point future prospective  Improve the English  Check  all structures are in a correct form Improve the conclusion section with the significant impact of these nanoformulations in the near future. Remove the table from the conclusion add it in earlier sections, where it fits better.

Author Response

We would like to thank the reviewer for the time, constructive feedback and recommendations.

Title has been changed.

Text corresponding to the point future prospective and the significant impact of these nanoformulations in the near future has been added to introduction and conclusion sections.

The manuscript has been carefully checked.

All structures were corrected and Figures 1, 3, 4, 8 were changed.

The Table 1 has been moved to the section 2. Links to the Table 1 were added to manuscript.